# Analysis of Genetic Diversity and Population Structure of Pigeonpea [*Cajanus cajan* (L.) Millsp] Accessions Using SSR Markers

**DOI:** 10.3390/plants9121643

**Published:** 2020-11-25

**Authors:** Didas Kimaro, Rob Melis, Julia Sibiya, Hussein Shimelis, Admire Shayanowako

**Affiliations:** 1African Centre for Crop Improvement, University of KwaZulu-Natal, Private Bag X01, Scottsville, Pietermaritzburg 3209, South Africa; melisr@ukzn.ac.za (R.M.); sibiyaj@ukzn.ac.za (J.S.); shimelish@ukzn.ac.za (H.S.); shayanowako@googlemail.com (A.S.); 2Dakawa Centre, Tanzania Agricultural Research Institute, P.O. Box, Morogoro 1892, Tanzania

**Keywords:** pigeonpea genetic diversity, population structure, polymorphism information content, molecular variance, SSR markers

## Abstract

Understanding the genetic diversity present amongst crop genotypes is an efficient utilization of germplasm for genetic improvement. The present study was aimed at evaluating genetic diversity and population structure of 48 pigeonpea genotypes from four populations collected from diverse sources. The 48 pigeonpea entries were genotyped using 33 simple sequence repeat (SSR) markers that are polymorphic to assess molecular genetic diversity and genetic relatedness. The informative marker combinations revealed a total of 155 alleles at 33 loci, with an average of 4.78 alleles detected per marker with the mean polymorphic information content (PIC) value of 0.46. Population structure analysis using model based revealed that the germplasm was grouped into two subpopulations. The analysis of molecular variance (AMOVA) revealed that 53.3% of genetic variation existed within individuals. Relatively low population differentiation was recorded amongst the test populations indicated by the mean fixation index (Fst) value of 0.032. The Tanzanian pigeonpea germplasm collection was grouped into three major clusters. The clustering pattern revealed a lack of relationship between geographic origin and genetic diversity. This study provides a foundation for the selection of parental material for genetic improvement.

## 1. Introduction

Pigeonpea (*Cajanus cajan* [L.] Millsp) is an important legume crop widely cultivated in Africa and Asia in diverse environments [1]. Pigeonpea belongs to the family *Leguminosae* and sub-family *Papilionoideae. Cajanus cajan* is the only domesticated species among *Cajanea* family. The crop has the ability to produce an economic yield under low moisture condition making it an important crop in dry areas [2]. Pigeonpea is one of the under-researched crops compared to other common legumes. In Tanzania, the crop is grown in several regions as a food and cash crop [3,4]. Tanzania is one of the top six global exporters of pigeonpea to the Asian market.

Pigeonpea is considered to be a valuable food and commercial crop with diverse uses. This crop improves soil physical properties and yield of associated intercropped crops while simultaneously yielding marketable grain. Their thick stems are mainly used as fuel food and roofing material [5,6]. Moreover, it provides an important source of protein for low-resource farmers who cannot afford animal protein [7]. In Australia, pigeonpea is used to reduce population of the cotton bollworm (*Helicoverpa armigera*) in “Bt” cotton production [8]. Furthermore, pigeonpea fixs atmospheric N more than any other legumes crop translating into savings of money that could have been used to purchase N fertilizers [9]. Pigeonpea acts as windbreak and contour hedges to control erosion [10].

Genetic diversity analysis in pigeonpea is crucial for effective breeding and germplasm conservation. Previous studies examined the genetic diversity of the crop using morphological and agronomic traits [11,12,13]. Also, recently DNA markers have been routinely used since they are readily available and not influenced by the environment [14]. However, the choice of markers to be used depends on the availability of genetic information about the genome sequence, cost for marker development, ease of documentation and polymorphism [15]. Among the DNA molecular markers, the simple sequence repeat (SSR) markers have been found to be the most suitable for use in pigeonpea [16]. A number of studies have been reported using SSR markers for detecting population structure and reconstructing the evolutionary history of the species [17,18,19]. The SSR markers are chosen because of high polymorphism, detection of multi-allelic variation, co-dominance, they are reproducible, they are easy to detect by polymerase chain reaction (PCR), are relatively abundant with a uniform coverage, require small amount of DNA and act as a universal genetic marker for the genetic reagent mapping [14,15,20]. Therefore, SSR markers are considered neutral markers. Cultivated pigeonpea are known to have low polymorphism, hence SSR markers are ideal for studying the genetic diversity [5,17]. Evaluating the genetic diversity and population structure of diverse pigeonpea germplasm collections is a key to deduce the genetic and population structure for breeding. Therefore, the aim of this study was to assess the genetic diversity and population structure present amongst 48 pigeonpea germplasm collections using SSR markers.

## 2. Materials and Methods

### 2.1. Plant Material

The study material consisted of 48 pigeonpea genotypes representing farmers’ landraces, landraces from the National Plant Genetic Resources Centre (NPGRC)-Arusha, Tanzania, breeding lines collected from International Crops Research for Semi-Arid Tropics (ICRISAT)-Nairobi, Kenya and released cultivars obtained from the Tanzania Agricultural Research Institute (TARI)-Ilonga, Morogoro-Tanzania. The selection of the plant material used relied on farmers preference for plant growth habit, resistant to *Fusarium* wilt and grain yield. The sample included the released pigeonpea varieties in Tanzania and most frequently used in breeding activities. The material used represent the collection from all major growing areas of pigeonpea in Tanzania. The detailed information about the material used in this study is given in Table 1.

### 2.2. DNA Extraction

The genomic DNA was extracted from fresh leaf material from 10- to 14 day-old-plants of each of the 48 pigeonpea entries and was ground to fine powder in liquid nitrogen, following the cetyl trimethyl ammonium bromide (CTAB) method as described by Mace et al. [21], with some modification. The quality and concentration of the extracted DNA was estimated by using a spectrophometer and the samples were diluted to a final concentration of 30 ng µL^−1^**.** For all samples, the DNA quality was determined by agarose gel electrophoresis (0.8% (w/v) stained with 5 µ/100 mL Gel^®^ (Biotium Inc., Hayward, CA, USA), while the quantity was determined by spectrophotometry (Nanodrop© 100, Thermo Scientific, Wilmington, DE, USA). The DNA samples were analysed at ICRISAT-laboratory, Centre of Excellence in Genomics, India.

### 2.3. Polymerase Chain Reaction (PCR) Amplification

Out of 35 SSR markers tested, only 33 SSR markers were polymorphic and used for this study (Table 2). The markers used for PCR amplification were selected based on sequence information and has been used in different genomic studies of pigeonpea [22]. The PCR amplification was optimized and conducted in a reaction buffer of 12.5 µL containing 1 × PCR buffer; 1 Unit Taq DNA polymerase; 0.2 mM each of dATP, dGTP and dTTP; 3 mM of MgCl_2_, 0.1 µM of respective forward and reverse primer and 40 ng of genomic DNA. The PCR amplification was carried out in a Bioer XP Thermal Cycler (Hangzhou Bioer Technologies, Hangzhou, China). The thermal cycling conditions were as follows: initial denaturation at 94 °C for 5 min, followed by 35 cycles of denaturation (94 °C) for 1 min, annealing (56–72 °C) for 1 min, primer extension (72 °C) for 1 min, followed by an extension at 72 °C for 20 min. The amplification products were analysed by electrophoresis on a 2.8% agarose gel, stained with ethidium bromide and photographed under short wavelength ultraviolet (UV) light in a gel documentation system. A 100 bp DNA ladder (MBI Fermentas, Baden Wurttemberg, Germany) was used as a size fragment standard.

### 2.4. Data Analysis

The allelic size was estimated visually for all 48 genotypes of pigeonpea. The band size of amplified products was determined by comparing with 100 bp DNA ladder (MBI Fermentas, Baden Wurttemberg, Germany). The SSR bands scored in pigeonpea genotypes was subjected to GenAlex version 6.5 [23] to assess the genetic diversity. Statistical parameters defining total number of alleles per locus (Na), number of effective alleles per locus (Ne), Shannon’s information index (I), observed heterozygosity (Ho), gene diversity (He), unbiased expected heterozygosity (uHe) and fixation index (F) were determined using the protocol of Nei and Li [24]. The polymorphic information content (PIC) values were calculated for each SSR locus as PIC = 1 − Σ (pi2), where pi is the frequency of the ith allele. An analysis of molecular variance (AMOVA) was performed to test the degree of differentiation among and within the sources of collection of the pigeonpea genotypes.

The population structure of the 48 pigeonpea was established using the Bayesian clustering method in STRUCTURE version 2.3.4 [25]. The length of the burn-in period and Markov Chain Monte Carlo (MCMC) were set at 10,000 iterations [26]. To obtain an accurate estimation of the number of populations, 20 n runs were performed for each K-value (assumed number of subpopulations), ranging from 1 to 10. Further, Delta K values were calculated and the appropriate K value was estimated by implementing the method of Evanno et al. [26] using the STRUCTURE Harvester program [27]. The relatedness was estimated by the genetic dissimilarity coefficients and the dendrogram were drawn using the unweighted pair group method (UPGMA) in DARwin 6.0 [28].

## 3. Results

### 3.1. Genetic Polymorphism of Simple Sequence Repeat (SSR) Markers

The sizes of amplified polymorphic DNA fragments (bands) ranged from 127 to 280 bp (Table 2). Genetic diversity parameters, such as number of alleles per locus (Na), number of effective alleles per locus (Ne), Shannon’s information index (I), observed (Ho) and expected (He) heterozygosity, unbiased expected heterozygosity (uHe), fixation index (F) and polymorphic information content (PIC) are presented in Table 3. A total of 158 alleles were amplified among the 48 pigeonpea and the numbers of alleles scored for 33 loci ranged from 2 to 11 with an average of 4.78. The lowest numbers of alleles per locus were detected from the markers CcM0484, CcM0594, CcM0673, CcM0785, CcM1045, CcM1251, CcM2379 and CcM2505. The maximum number of alleles (11) was detected at the CcM0246 and CcM0443 locus. The markers CcM2379 and CcM0246 had the lowest and highest numbers of effective alleles of 1.02 and 6.14, respectively.

The PIC value of the SSR markers, which is a measure of allele diversity at a locus, ranged from 0.04 to 0.84 with an average of 0.44. The observed variations in PIC value in this study could be attributed to genotypic differences in the pigeonpea material used. Eight SSR loci (CcM0246, CcM0381, CcM0443, CcM0492, CcM0721, CcM0974, CcM2704 and CcM2895) exhibited PIC values higher than 0.70 indicating their usefulness in discriminating genotypes. The observed heterozygosity values ranged from 0.0 to 0.69 with an average of 0.26. The gene diversity ranged from 0.04 to 0.85, with a mean of 0.45 (Table 3). Markers CcM2379 and CcM0246 had the lowest and highest genetic diversities, respectively. The inbreeding coefficient (F_IS_) ranged from 0.03 to 0.93 with a mean of 0.39. Two loci (CcM0698 and CcM2379) showed an F_IS_ value of 1 suggesting the alleles at these loci are fixed i.e., reached 100% homozygosity and, therefore were excluded from the analysis. However, six loci (CcM0484, CcM0444, CcM2004, CcM2044, CcM2505 and CcM2049) showed a very low F_IS_ values signifying that the alleles at these loci are heterozygotes.

### 3.2. Genetic Relationship among 48 Pigeonpea Genotypes Based on Source of Collection

The genetic parameters of the population studied based on source of collection were presented in Table 4. Among the four-population investigated, the mean values of observed alleles (Na) and effective alleles (Ne) were 3.52 and 2.30, respectively. Popn 1 (Improved genotypes-ICRISAT) recorded the lowest values of Na (3.27) and Ne (2.10). Similarly, the highest values of Na (4.15) and Ne (2.47) were recorded from popn 4 (landraces-NPGRC). Shannon’s information index ranged from 0.74 to 0.92 with a mean of 0.83. The observed heterozygosity (Ho) ranged from 0.19 (popn 1) to 0.35 (popn 2-Improved genotypes -TARI) and expected heterozygosity (He) from 0.39 (popn 1) to 0.47 (popn 4), with an average of 0.28 and 0.43, respectively. The unbiased expected heterozygosity ranged from 0.41 (popn 1) to 0.49 (popn 2) with an average of 0.46. The values for inbreeding coefficient ranged from 0.16 (popn 2) to 0. 40 (popn 1) with an average of 0.28 at the population level.

### 3.3. Population Structure of 48 Pigeonpea Accessions

Based on the ∆K value, the population structure analysis of the 48 pigeonpea genotypes grouped the population into two subpopulations (Figure 1). Similarly, the maximum ad hoc measure ∆K was found to be K = 2 (Figure 1). Membership of all genotypes to a particular sub-cluster was based on at least 70% ancestry. Cluster 2 had the largest membership with 78% of the population, while the smallest was Cluster 1 with only 22% (Table 5). Sub-population 1 comprised of genotypes from landraces (54.5%) and improved cultivars (45.5%) mainly long maturity duration, and sub-population 2 consisted of genotypes from landraces (64.9%) and improved cultivars (35.1%) mainly medium-maturity duration.

### 3.4. Genetic Cluster Analysis of 48 Pigeonpea Accessions

The UPGMA cluster analysis based on genetic dissimilarity using the neighbour-joining method in DARwin 5.0 grouped the 48 pigeonpea genotypes into three genetic clusters (Figure 2). Three genetic clusters identified were not consistent with the results of structure analysis. There is no other cluster made up exclusively of accession from the same geographic location except cluster I. Cluster I contained only 1 accession (landraces from NPGRC). Cluster 2 contained 17 accessions (7 landraces from NPGRC, 3 landraces from farmers, 4 released cultivars from TARI and 3 breeding line from ICRISAT). Cluster III contained 30 accessions (9 breeding line from ICRISAT, 12 landraces from NPGRC, 3 released cultivar from TARI and 6 landraces from farmers). Genotypes with the most distinct genetic make-up are ICEAP 00040, ICEAP 00932, Bangili, Babati White, ICEAP 00557, ICEAP 00554, TZA 253, TZA 5596 and TZA 2466 could be considered for future breeding.

### 3.5. Analysis of Molecular Variance (AMOVA)

AMOVA was performed in model-based populations (Table 6). The results of AMOVA revealed that the majority of variance occurred within individuals and accounted for 53.3% of the total variation, whereas 3.2% and 43.5% of the variation was attributed to differences between population and among individuals. Calculation of Wright’s [29] F statistic at all SSR loci revealed that F_IS_ was and 0.449 and F_IT_ was 0.466. Determination of mean fixation index (F_ST_) for the polymorphic loci across all accessions indicated that F_ST_ was 0.0318 which implies low genetic variation across genetic subgroups. The haploid Nm was very high (7.6) indicating a high gene exchange among populations (Table 6). These results demonstrated that genetic differentiation among subpopulations was low and within subpopulations was high.

## 4. Discussion

The estimated average PIC value (0.46) recorded in the current study was similar to PIC value (0.49) reported by Sousa et al. [18], lower than a mean PIC (0.57) reported by Zavinon et al. [19] but relatively higher than the average PIC value of 0.37 as reported by Bohra et al. [30]. Similarly, Singh et al. [31] observed a PIC value of 0.515. Also, Singh et al. [32] using SSR markers to assess diversity of 40 genotypes, including four wild relatives observed a high mean PIC value of 0.523. However, a PIC value obtained from this study is greater than the study reported by Njung’e et al. [17] on pigeonpea. Low PIC values of 0.18 and a high PIC value of 0.90 was reported by Walunjkar et al. [33] using SSR and random amplified polymorphic DNA (RAPD) markers, respectively. Rani et al. [34] using 10 RAPD and 10 ISSR markers for 42 genotypes reported high PIC values of 0.73 and 0.77, respectively. Using genotyping-by-sequencing (GBS) and single nucleotide polymorphisms (SNPs) a lower PIC value of 0.25 was reported [19]. The low levels of polymorphism reported in this study are in agreement with previous findings of several researchers [35,36,37], who reported low polymorphism in cultivated pigeonpea genotypes. Therefore, the SSR markers used in this study confirmed the existence of genetic variation in pigeonpea germplasm.

Genetic improvement of crops depends on the amount of genetic variation among the breeding material. Previous studies of genetic diversity in Brazilian pigeonpea genotypes by Sousa et al. [18] and Malawian germplasm by Njung’e et al. [17], using SSR markers reported a higher average number of alleles of 5.1 and 5.58, respectively. An earlier study of Kenyan pigeonpea and Indian accessions by Songok et al. [38] using 88 genotypes and six SSR markers reported a high average number of alleles per locus. This shows that genotypes studied in Kenya, Malawi and Brazil have a higher diversity than that of Tanzania. The reason for low diversity of pigeonpea in Tanzania as revealed by SSR markers is attributed due to low number of genotypes used in this study and lack of wild relatives conserved in the gene bank. The low average number of alleles observed in this study could be attributed to the fact that the genotypes were collected from a relative narrow geographical area.

The average gene diversity (*H**_e_*), which is a measure of genetic diversity observed, in the present study, was low compared to most previous studies [38] which reported a higher genetic diversity in Indian genotypes than East African germplasm. Wasike et al. [39] investigated the Asian and African accession using AFLP markers and revealed that Indian accession are more diverse than African accession. These findings and several other reports suggest that India is the centre of origin and domestication of pigeonpea and East Africa is the secondary centre of genetic diversity. In the present study, the observed heterozygosity of the genotypes was low. Comparable results were obtained from a microsatellite-based study that involved 77 accessions from Brazil using 43 SSR markers [18]. The low level of observed heterozygosity is mostly likely attributable to the farmers selection pressure that might have reduced polymorphism in the populations.

The population structure analysis based on STRUCTURE revealed the presence of two subpopulations among the 48 pigeonpea accessions collected from a subset of Tanzanian and Kenyan germplasm. This result is supporting the earlier findings by Bohra et al. [30] who reported two subpopulations among 94 tested pigeonpea. By contrast, population structure of South American pigeonpea accessions by Sousa et al. [18] reported four subpopulations.

In the present study, the genetic variation components confirmed that there is fair genetic diversity among individuals within the population (43.5%) than among populations (3.2%). The percentage variation within groups was as high as 53.3% of the given populations. The current study is in agreement with the findings of previous publications on pigeonpea [40,41], which reported a higher percentage variation within groups than among populations and among individuals. In a similar way, Bohra et al. [30] reported a high genetic diversity within the population (89%) than among individuals (11%). The value of Fst was observed to be 0.031, indicating little differentiation among populations. The fixation index (Fst) obtained in the current study were lower than those of Kassa et al. [42] and Kumar et al. [40] with 0.949 and 0.17, respectively.

The dendrogram based on SSR markers revealed three major clusters. This indicates the existence of a high degree of genetic diversity in the germplasm evaluated in this study. Therefore, these germplasms would serve as a valuable source for the selection of diverse parents useful for plant breeders to improve the existing commercial cultivars. In this study distinct clustering was not observed according to geographical basis or on the basis of maturity duration. This was also reported by Petchiammal et al. [43], who recorded 21 pigeonpea genotypes including 3 wild relatives with the same morphological features grouped into different clusters. By contrast, Singh et al. [44] and Songok et al. [38] reported a grouping of pigeonpea genotypes on geographical basis. Also, Bohra et al. [30] reported a pattern of grouping genotypes of long maturity duration within the same cluster similar to the previous findings [31,45].

## 5. Conclusions

This study deduces the presence of a considerable level of genetic diversity among pigeonpea genotypes in Tanzania. This will serve as basic information by providing options to breeders to develop, through selection and breeding, new and more productive varieties that are adapted to changing environments. This germplasm could also be used for mapping population studies, producing a core collection, and facilitates the identification of useful traits. Additionally, with the enlarged sampling area, the existing germplasm will allow us to search for additional rare alleles.

## Figures and Tables

**Figure 1 plants-09-01643-f001:**
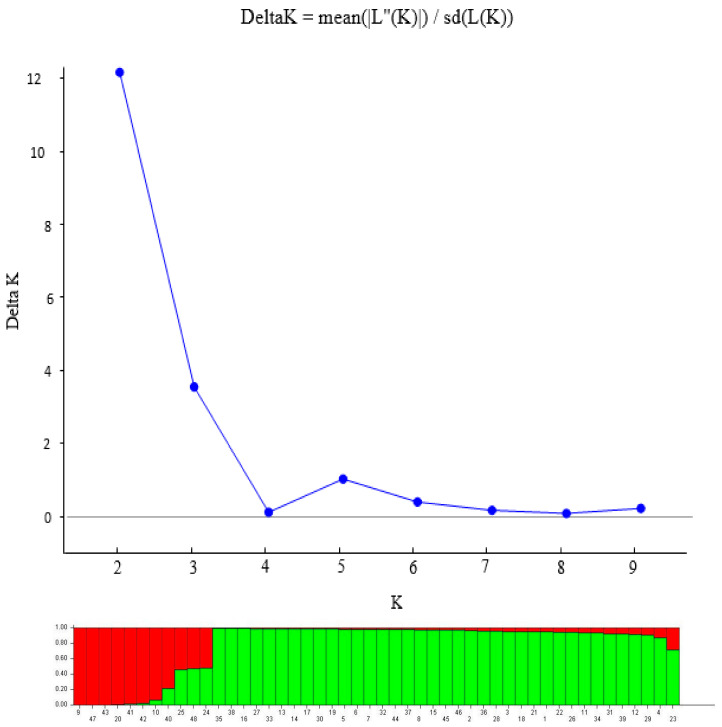
Population structure of 48 accessions based on 35 SSR markers (K = 2) and graph of estimated membership fraction for K = 2. The maximum of ad hoc measure ΔK determined by structure harvester was found to be K = 2, which indicated that the four populations can be grouped into two subgroups.

**Figure 2 plants-09-01643-f002:**
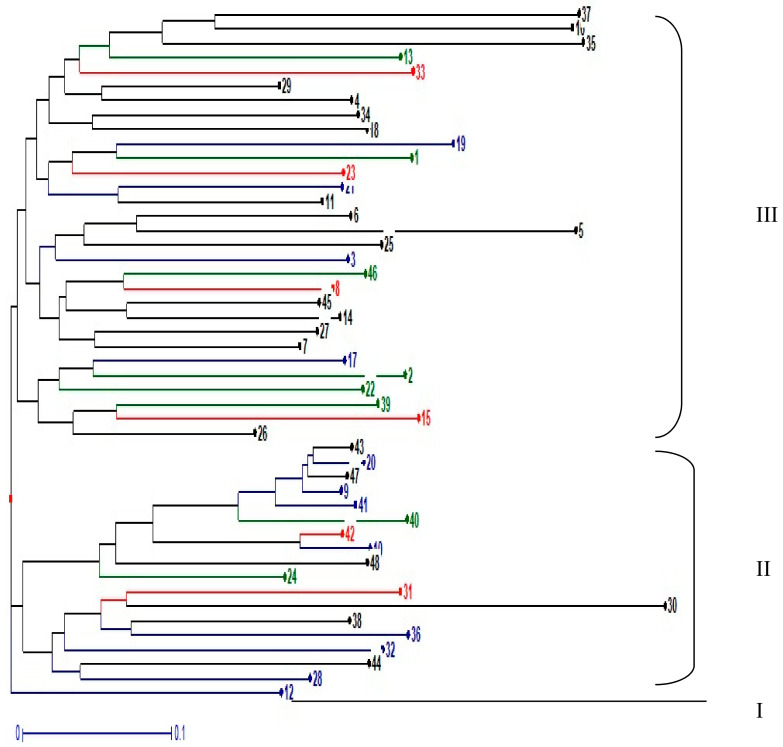
Dendrogram showing genetic relationship among 48 pigeonpea genotypes tested using 35 SSR markers. Accessions with the same colour share the same source of collection.

**Table 1 plants-09-01643-t001:** List of the pigeonpea genotypes evaluated in this study, their place of collection and country of origin.

S/N	Genotype	Status	Origin	Country	S/N	Genotype	Status	Origin	Country
1	ICEAP 00040	Released cultivar	TARI	Tanzania	25	ICEAP 01172/2	Breeding line	ICRISAT	Kenya
2	ICEAP 00936	Breeding line	ICRISAT	Kenya	26	ICEAP 01154/15	Breeding line	ICRISAT	Kenya
3	Babati White	Landraces	Farmers	Tanzania	27	ICEAP 00979/1	Breeding line	ICRISAT	Kenya
4	TZA 253	Landraces	NPGRC	Tanzania	28	ICEAP 01172/2	Breeding line	ICRISAT	Kenya
5	TZA 2466	landraces	NPGRC	Tanzania	29	ICEAP 01154/2	Breeding line	ICRISAT	Kenya
6	TZA 2785	Landraces	NPGRC	Tanzania	30	ICEAP 0673-1	Breeding line	ICRISAT	Kenya
7	TZA 197	landraces	NPGRC	Tanzania	31	ICEAP 00554	Released cultivar	TARI	Tanzania
8	ICEAP 00557	Released cultivar	TARI	Tanzania	32	Mthawanjuni	Landraces	Farmers	Tanzania
9	ICEAP O1179	Breeding line	ICRISAT	Kenya	33	TZA 2514	Landraces	NPGRC	Tanzania
10	Bangili	Landraces	Farmers	Tanzania	34	TZA 2464	Landraces	NPGRC	Tanzania
11	ICEAP 00540	Breeding line	ICRISAT	Kenya	35	TZA 5596	Landraces	NPGRC	Tanzania
12	Kondoa	Landraces	Farmers	Tanzania	36	TZA 5582	Landraces	NPGRC	Tanzania
13	TZA 2692	Landraces	NPGRC	Tanzania	37	Arumeru	landraces	Farmers	Tanzania
14	No. 40	Landraces	Farmers	Tanzania	38	TZA 5463	Landraces	NPGRC	Tanzania
15	ICEAP 00911	Breeding line	ICRISAT	Kenya	39	TZA 2496	Landraces	NPGRC	Tanzania
16	TZA 250	Landraces	NPGRC	Tanzania	40	Tumia	Released cultivar	TARI	Tanzania
17	TZA 5464	Landraces	NPGRC	Tanzania	41	ICEAP 00932	Released cultivar	TARI	Tanzania
18	Komboa	Released cultivar	TARI	Tanzania	42	Ilonga	Landraces	Farmers	Tanzania
19	TZA 5557	Landraces	NPGRC	Tanzania	43	ICEAP 00576-1	Breeding line	ICRISAT	Kenya
20	TZA 2509	Landraces	NPGRC	Tanzania	44	Hombolo	Landraces	Farmers	Tanzania
21	Kiteto	Landraces	Farmers	Tanzania	45	TZA 5541	Landraces	NPGRC	Tanzania
22	TZA 5555	Landraces	NPGRC	Tanzania	46	TZA 2456	Landraces	NPGRC	Tanzania
23	ICEAP 01147	Breeding line	ICRISAT	Kenya	47	TZA 2807	Landraces	NPGRC	Tanzania
24	TZA 2439	Landraces	NPGRC	Tanzania	48	ICEAP 00053	Released cultivar	TARI	Tanzania

NB: TARI-Tanzania Agricultural Research institute; ICRISAT—International Crops Research for Semi-Arid Tropics; NPGRC—National Plant Genetic Resources Centre.

**Table 2 plants-09-01643-t002:** Primer sequences of the 35 simple sequence repeat (SSR) markers used in this study.

Marker Name	GenBank ID	SSR Motif	Forward Primer (5′–3′)	Reverse Primer (5′–3′)	Product Size (bp)
CcM0121	FI191501	(TA)17	AGAAATTGGAGGCTTGGTCA	GGTATAAGGCTCAAACCCGA	273
CcM0443	FI200654	(TA)17n(AT)5	TGACAAAATAATGCGGTCACA	CAAGCCAAAGTTTGTTTGAACT	261
CcM2044	FI245729	(TAT)9	ATCACTCCAAGCACCCAAAC	TGCAAATGGAAGGGAATAGC	212
CcM0444	FI200657	(TA)7	TGTCATGAGTGGCTGATCCT	TCAACCAAAATCCAAACCAA	184
CcM0494	FI202253	(AT)21	ACGTGAAAAATCCGCAACTT	GCTTGTGTTTCAAAATCCAACTT	117
CcM1251	FI224872	(CCA)9	CAAATGGCAGAACAGAGCAG	CGGAGATTGCATTGTTCCTT	228
CcM2097	FI246959	(CT)12	TGATAGGAATATTTCGGCGG	CCTTTGAAATTGAAGGCGAG	193
CcM2409	FI255641	(TTA)6	TGAAGGTTGATCCAAGGAGG	CGTGCAAAATAATTGTCCAAAA	186
CcM0974	FI216621	(AT)13	CGTCTTACAGACGATCTGCATC	CAAAGAAACAGACATGATAAAGAGAGA	161
CcM2379	FI254391	(TC)10	CCGGAAAAATTGCCTATTGA	TTCGATGACAGAATTTAGGTGC	151
CcM2895	FI272645	(AT)24	AATGATAATTGGACACTTCTTTTTC	TGCGTTAATTAAACAAGCAAGC	268
CcM0246	FI195094	(AT)16	ATGGAGCCAAAGTGTCCAAG	ATGAAAAGCAACTACGCGCT	226
CcM0721	FI209310	(AT)19	ATCCAACCACGTGTTTCACA	TTTGAAATGGTATCGATGATTAAA	169
CcM0195	FI193462	(AT)11	CAACAATAAAGCATAAACCACCA	TGACGTAGATTGGGTAGTTAGGA	223
CcM0361	FI198648	(TA)9	TCTTCCTGTCCTCATCCTCG	TGGAAACCAAAGTTGTGCAT	172
CcM0374	FI198903	(TA)11	GAACCGTCTTAAAATTTCTCATTT	CAATGGCACATTGTCAAAAA	161
CcM0484	FI201979	(T)12n(ATT)5n(AT)5	TGGAAATTAAACACCATGAAACA	TGCATGCTACCAAGGAATTG	248
CcM2049	FI245893	(TAT)9	GCGACCAGGTACTTTCAAGC	CGAAAAGCGATTTCAGAATTT	260
CcM0594	FI205393	(GA)9n(TC)9	GGCTTGGTTCTTTCTTGGTG	AAGTCCCTGACTTTCCCCAT	185
CcM0956	FI216271	(AT)16	AGCCCCAACTCAATTATCAAA	TTCCTTGCGGTTTGAGCTAT	224
CcM1357	FI227664	(AT)15n(ATA)5	TCTAGCATCTCCATTAAACCATTT	ACACATATGACATTTAGCAAATAAAAA	280
CcM2004	FI244896	(CT)7n(AG)12	AGGAATGCGACATTTTGGAG	TCCCCATCCCTTTCTTTCTT	209
CcM0492	FI202198	(AT)21	AAAATTTACGAGCACTAAAATGAAAAA	TCAACAATAAATTGTCATATGTCTGG	271
CcM0698	FI208758	(AAT)17	CTCTTCTTGTTGTCCCTCGC	GCAGTTCTGGAATACCTCGC	188
CcM0785	FI210851	(AT)9	GCATGTGTTTTTACTTGAGTCGTC	TGGAGGCGATCTCTTTCTTG	277
CcM1982	FI244391	(TC)17	TATCAAACCTGGCGATCACA	ATTCCGCAAACACATCACAA	246
CcM2704	FI265930	(AT)10	AAAAATGTTCAATGTCGTAGTATTTGA	TGCCATATATCATGCCCTCA	127
CcM0248	FI195265	(TA)8	CAAACTCAACCCTACCAATGC	CATTCCTTGTCATCAATGAAGTTT	280
CcM0673	FI208212	(AT)6(AG)9	TGACCACCAACCATTACCAA	CATGCACCAGACCAGAATCA	272
CcM1045	FI219229	(AT)6	AACCTTAGTTGGTGATAGATTTCAGA	ACCGTCAAGTCCCAAATCAC	262
CcM2394	FI255036	(TC)12	TGGAAACGATTTCCTACCACA	ACAAGGGGAAAAGGGAAAGA	260
CcM0381	FI199172	(TA)21	CGATCCCTGCTTGAAATCAT	GGTTCAAGCGATGCACTACA	267
CcM0834	FI212739	(AT)10	GTCCGGCTTGCCTATAAGGT	AAGGCAACCTCCCCAGTATT	262
CcM2505	FI259241	(GAA)8	CCTCGGAAGAGATTGCAGTT	TGATGAATTGGGAAGCAACA	201
CcM2697	FI265781	(CT)9n(T)14	AGAGTTCGGTGACGGTTACG	GATCTGTCGAGGTTGAGGCT	242

Source: [22].

**Table 3 plants-09-01643-t003:** Genetic parameters generated by 33 SSR markers on 48 pigeonpea genotypes.

Markers	N	Na	Ne	I	Ho	He	uHe	F	Pic
CcM0121	47	5	2.03	1.01	0.21	0.51	0.51	0.58	0.51
CcM0195	48	4	3.01	1.19	0.44	0.67	0.67	0.34	0.67
CcM0246	47	11	6.15	2.03	0.36	0.84	0.85	0.57	0.84
CcM0248	47	4	1.68	0.80	0.32	0.40	0.41	0.21	0.40
CcM0361	44	4	2.79	1.20	0.05	0.64	0.65	0.93	0.64
CcM0374	48	3	1.18	0.33	0.08	0.16	0.16	0.46	0.16
CcM0381	46	7	4.06	1.60	0.26	0.75	0.76	0.65	0.75
CcM0443	48	11	5.48	1.91	0.42	0.82	0.83	0.49	0.82
CcM0444	48	4	1.11	0.25	0.10	0.10	0.10	−0.04	0.10
CcM0484	47	2	1.24	0.34	0.17	0.19	0.19	0.11	0.19
CcM0492	46	9	5.17	1.83	0.46	0.81	0.82	0.43	0.81
CcM0494	48	6	2.89	1.34	0.29	0.65	0.66	0.55	0.65
CcM0594	46	2	1.24	0.34	0.13	0.19	0.20	0.33	0.19
CcM0673	47	2	1.94	0.68	0.28	0.49	0.49	0.43	0.49
CcM0721	46	6	3.49	1.48	0.50	0.71	0.72	0.30	0.71
CcM0785	47	2	1.26	0.36	0.15	0.21	0.21	0.28	0.21
CcM0834	48	4	1.29	0.48	0.17	0.23	0.23	0.26	0.23
CcM0956	39	6	2.91	1.33	0.31	0.66	0.67	0.53	0.66
CcM0974	47	8	3.78	1.63	0.49	0.74	0.74	0.34	0.74
CcM1045	48	2	1.16	0.26	0.10	0.14	0.14	0.23	0.14
CcM1251	39	2	1.85	0.65	0.26	0.46	0.47	0.44	0.46
CcM1357	46	5	2.07	0.96	0.37	0.52	0.52	0.29	0.52
CcM1982	47	4	1.30	0.52	0.17	0.23	0.24	0.27	0.23
CcM2004	47	3	1.16	0.30	0.15	0.14	0.14	−0.07	0.14
CcM2044	48	4	2.99	1.17	0.69	0.67	0.67	−0.03	0.67
CcM2049	47	3	1.30	0.47	0.21	0.23	0.23	0.08	0.23
CcM2097	46	5	2.37	1.07	0.37	0.58	0.59	0.36	0.58
CcM2394	47	3	1.11	0.23	0.06	0.10	0.10	0.37	0.10
CcM2409	48	3	1.53	0.57	0.19	0.35	0.35	0.46	0.35
CcM2505	47	3	1.09	0.20	0.09	0.08	0.08	−0.04	0.08
CcM2697	48	4	2.22	0.95	0.31	0.55	0.56	0.43	0.55
CcM2704	46	10	4.52	1.73	0.50	0.78	0.79	0.36	0.78
CcM2895	47	7	4.62	1.67	0.47	0.78	0.79	0.40	0.78
Mean	46.52	4.60	2.48	0.94	0.28	0.47	0.47	0.34	0.47
S.E	0.37	0.45	0.25	0.10	0.03	0.04	0.05	0.04	

Na = Number of different alleles; Ne = number of effective alleles = 1/(Sum pi^2); I = Shannon’s Information Index = −1 × Sum (pi × Ln (pi)); Ho = observed heterozygosity = No. of Hets/N; He = expected heterozygosity = 1 − Sum pi^2; uHe = Unbiased expected heterozygosity = (2N (2N − 1)) × He; F = Fixation index = (He − Ho)/He = 1 − (Ho/He); PIC = Polymorphic information content; SE = Standard error; where Pi = is the frequency of the ith allele for the population & Sum pi^2 is the sum of the squared population allele frequencies.

**Table 4 plants-09-01643-t004:** Summary of different *C. cajan* population diversity statistics averaged over the 35 SSR loci.

Population	N	Na	Ne	I	Ho	He	uHe	F
Popn 1	11.58	3.27	2.10	0.75	0.20	0.40	0.42	0.41
Popn 2	5.88	3.24	2.34	0.83	0.35	0.45	0.49	0.16
Popn 3	8.79	3.42	2.30	0.82	0.28	0.43	0.46	0.27
Popn 4	20.27	4.15	2.47	0.93	0.30	0.48	0.49	0.31
Mean	11.63	3.52	2.30	0.83	0.28	0.44	0.46	0.29
SE	0.47	0.16	0.11	0.05	0.02	0.02	0.02	0.03

Popn1 = Improved-ICRISAT, Popn2 = Improved-TARI, Popn3 = Landraces Farmers, Popn4 = Landraces NPGRC, Na = Number of different alleles; Ne = number of effective alleles, I = Shannon’s Information Index, Ho = observed heterozygosity, He = expected heterozygosity, uHe = Unbiased expected heterozygosity, F = Fixation index.

**Table 5 plants-09-01643-t005:** Genetic clusters and member of genotypes observed from population structure analysis of 48 pigeonpea genotypes.

Cluster	Genotype	% Membership	He	Fst
1	E9, E10, E20, E40, E40, E41, E42, E43, E47	22	0.16	0.71
2	E1, E2, E3, E4, E5, E6, E7, E8, E9, E12, E13, E13, E14, E15, E16, E17, E18, E19, E21, E23, E26, E27, E30, E31, E32, E33, E34, E35, E36, E37, 38, E39, E44, E46	78	0.54	0.01

He = expected heterozygosity; Fst = mean fixation index.

**Table 6 plants-09-01643-t006:** Analysis of molecular variance among and within the 48 pigeonpea genotypes.

Source	df	SS	MS	Est. Var.	%	F-Statistics
Among Populations	3	52.76915	17.58972	0.263732	3.2	0.001
Among Individual	44	511.2413	11.61912	3.601227	43.5	0.001
Within Individual	48	212	4.416667	4.416667	53.3	0.001
Total	95	776.0104		8.219584	100	
Fixation indices	Value					
Fst	0.031845					
Fis	0.449149					
Fit	0.466691					
Nm	7.60042					

df = degree of freedom; SS = Sum of squares; Est. Var = Estimated variance; % = percent variation; Fst = Fixation index; Fis = Inbreeding coefficient; Fit = Overall fixation index.

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
