# Peer review of "Analysis of Genetic Diversity and Population Structure of Pigeonpea [Cajanus cajan (L.) Millsp] Accessions Using SSR Markers"

_plants, 2020, doi:10.3390/plants9121643_

Round 1
Reviewer 1 Report
Dear Authors,
Please find my comments and suggestions about your study submitted to the 'Plants' journal.
Briefly, I appreciated to read it but I am not able to appreciate the interest for the scientific field. Nevertheless, I considered the methods employed and the results reported are strongly supported by other studies and well discussed. However, minor adjustements have to be made to reach publication standards. Notably, i found that offering a perspective of the use of such results in the plant breeding context would benefit to the quality of the present MS. For other comments, please find these included in the revised PDF.
Thus I strongly encorage authors to produce a revised version of their MS
All the best

Reviewer 2 Report
In this study, Kimaro et al. studied genetic diversity and population structure of 49 pigeon pea genotypes using 33 simple sequence repeat (SSR) markers. In general, the purposes and experimental approaches of manuscript are fine for publication. However, there are several errors in the manuscript. The manuscript does not contain page and line numbers. The figures and tables should be imbedded within the results section. It is desirable to check the manuscript by a professional English editing service. Overall, I recommend this manuscript for publication after major revision. My comments are as follows.
Title
(Cajanus cajan (L.) Millsp -> [Cajanus cajan (L.) Millsp]
Abstract:Understanding -> Abstract: Understanding
Introduction
(Cajanus cajan (L.) Millsp) -> [Cajanus cajan (L.) Millsp] In addition, please include information of the family for pigeonpea.
References should be reformatted according to Plants. Check them.
Bt cotton -> Spell out “Bt”
pigeonpea fix atmospheric N more -> pigeonpea fixs atmospheric N more
Introduction is written well. But many sentences are started from “It”. It might be desirable to write them differently.
Materials and methods
Gel® (Biotium Inc., USA) -> Gel (Biotium Inc., City, U.S.A.) Name of city should be included.
(Nanodrop© 100, Thermo Scientific, USA) -> (Nanodrop 100, Thermo Scientific, USA). Name of city should be included.
In 2.3. PCR amplification, please provide the detail of selection for SSR makers in detail.
MgCl2 -> MgCl2 (2 should be subscript).
40ng -> 40 ng
oC -> Check degree sign °C
(MBI Fermentas, Germany) -> City should be included.
Data analysis section is nicely written.
Results
280bp -> 280 bp
Results section are nicely written.
Discussion
Using GBS derived SNPs -> Spell out GBS and SNPs
Round 2
Reviewer 2 Report
Authors revised the manuscript properly according to reviewer's comments.